# Multi-omics investigation of *Porphyromonas gingivalis* exacerbating acute kidney injury through the gut-kidney axis

Ling Dong,[1] Zhaoxin Ji,[1] Jing Sun,[2] Jiangqi Hu,[1] Qingsong Jiang,[1] Wei Wei[1]

**ABSTRACT** Periodontitis is closely related to renal health, but the specific influence of *Porphyromonas gingivalis* (*P. gingivalis*), a key pathogen in periodontitis, on the development of acute kidney injury (AKI) in mice has not been fully elucidated. In our study, AKI was induced in mice through ischemia-reperfusion injury while administering oral infection with *P. gingivalis*. Comprehensive analyses were conducted, including 16S rRNA sequencing, liquid chromatography-mass spectrometry (LC-MS) metabolomics, and transcriptome sequencing. *In vitro*, the identified metabolite was used to stimulate mouse neutrophils. Subsequently, these modified neutrophils were co-cultured with mouse renal tubular epithelial cells. The results showed that oral infection with *P. gingivalis* significantly exacerbated AKI in mice. 16S rRNA sequencing revealed notable shifts in gut microbiota composition. LC-MS metabolomics analysis identified significant metabolic alterations, particularly the upregulation of 3-indoleacrylic acid in the serum. Transcriptome sequencing showed an increased expression of neutrophilic granule protein (*Ngp*), which was closely associated with 3-indoleacrylic acid, and the presence of *Porphyromonas*. Cellular experiments demonstrated that 3-indoleacrylic acid could activate neutrophils, leading to an elevation in NGP protein levels, a response that was associated with renal epithelial cell injury. Oral infection with *P. gingivalis* exacerbated AKI through the gut-kidney axis, involving gut microbiota dysbiosis, metabolic disturbances, and increased renal expression of *Ngp*.

**IMPORTANCE** This study provides novel insights into the relationship between periodontal health and renal function. *Porphyromonas gingivalis* oral infection disrupted the balance of gut microbiota and was an important modifier determining the severity of acute kidney injury. Under the "gut-kidney axis," *P. gingivalis* might cause an increase in the level of the gut microbial metabolite 3-indoleacrylic acid, interfering with kidney immunity and disrupting the maintenance of kidney epithelium.

**KEYWORDS** acute kidney injury, *Porphyromonas gingivalis*, gut-kidney axis, gut microbiota, metabolomics

Acute kidney injury (AKI) is a clinical syndrome characterized by a rapid decline in renal function. AKI encompasses a variety of pathophysiological processes, including inflammation, oxidative stress, epigenetic changes, and other mechanisms (1). Persistent renal dysfunction is associated with irreversible damage to renal cells and nephrons, potentially progressing to chronic kidney disease (2). In recent years, significant progress has been made in understanding the critical role of crosstalk between the kidneys and distant organs in AKI (3, 4).

Epidemiological evidence indicates that patients with periodontitis exhibit significant renal function impairment compared to the general population (5), underscoring the importance of considering periodontitis as closely related to renal health. *Porphyromonas gingivalis* (*P. gingivalis*) is a major pathogen in periodontitis, known to cause the

**Peer Reviewer** Ran Zhang, Department of Oral Pathology, Peking University School and Hospital of Stomatology, Beijing, China

Address correspondence to Wei Wei, hxkqww@163.com.

The authors declare no conflict of interest.

See the funding table on p. 13.

destruction of periodontal supporting tissues (6, 7). Our previous studies have also confirmed that oral infection with *P. gingivalis* exacerbates renal damage in AKI in mice (8). However, the specific mechanisms by which orally derived *P. gingivalis* damages distant renal tissues remains unclear.

The gut microbiota is a complex ecosystem within the human intestine that continuously communicates with key host organ systems to regulate health (9). Recently, the "gut-kidney axis" theory has provided a new perspective on the bidirectional communication between the gut microbiota and kidney disease (10). On the one hand, impaired renal function can lead to gut microbiota dysbiosis (11, 12); on the other hand, alterations in the gut microbiota may damage the intestinal mucosal barrier, allowing harmful bacteria to enter the bloodstream, trigger inflammatory responses, and accelerate kidney damage (13). Additionally, an increase in nephrotoxic metabolites produced by dysregulated gut microbiota within the "gut-kidney axis" is another mechanism that exacerbates kidney disease (14). Oral infection with *P. gingivalis* has been shown to disrupt the composition and function of the gut microbiota (15, 16), not only disturbing the intestinal environment but also subsequently leading to extensive extra-intestinal physiological reactions, such as liver disease and autoimmune disorders (17). Nevertheless, there is a lack of compelling evidence to support the notion that oral infection with *P. gingivalis* exacerbates AKI by disrupting the gut microbiota.

This study aims to investigate the impact of oral infection with *P. gingivalis* on the gut microbiota and metabolism in a mouse model of AKI using multi-omics approaches. We seek to elucidate the direct pathophysiological role of gut microbiota dysbiosis induced by *P. gingivalis* in AKI.

## MATERIALS AND METHODS

### Experimental mouse model

Male C57BL/6J mice, aged 8 weeks, were randomly assigned to various experimental groups. The study protocols were approved by the Institutional Animal Care and Use Committee of Capital Medical University.

To induce AKI, a dorsal skin incision was made to expose both kidneys, and the renal pedicles were clamped for 30 minutes. In the sham group, mice underwent anesthesia and muscle incision without clamping (18). The surgical procedures were performed on the 55th day following the initial bacterial inoculation. The C57BL/6 strain exhibited a pronounced AKI phenotype following surgery, making it a highly suitable model for our study. Using this well-characterized strain ensures reliable results and facilitates comparisons with prior research in the field.

The *P. gingivalis* W83 bacteria were centrifuged to form a pellet, which was then resuspended in sterile 3% (wt/vol) carboxymethylcellulose to achieve a concentration of $1 \times 10^{10}$ CFU/mL. A 100 µL aliquot of this suspension was orally administered to the mice every 2 days for a total of eight inoculations (19, 20).

Antibiotic treatment was administered by providing autoclaved drinking water containing 1 g/L ampicillin (Solarbio, Cat#A8180-1), 1 g/L metronidazole (MCE, Cat#HY-B0318), 1 g/L neomycin (MCE, Cat#HY-B0470), and 0.5 g/L vancomycin (MCE, Cat#HY-B0671) (21). This treatment started 7 days before the AKI surgery and continued throughout the study.

Samples were collected on the first day post-surgery. Under sterile conditions, fecal samples were obtained from the mice and stored at −80℃. The mice were weighed, and blood samples were collected. After euthanasia by $CO_2$ asphyxiation, the kidneys were harvested and weighed. Serum was obtained from whole blood samples for renal function testing. The experimental workflow is illustrated in Fig. S1.

## HE staining and Masson's trichrome staining

Kidney and intestinal tissues were embedded in paraffin. HE staining was performed according to the kit's instructions (Servicebio, Cat#G1076). Masson's trichrome staining was conducted as per the manufacturer's guidelines (Servicebio, Cat#G1006).

## Immunofluorescence staining

Samples were pretreated with trypsin and permeabilized using 0.5% PBST. They were then incubated with hydrogen peroxide solution and blocked with goat serum. Primary antibodies specific for TLR4 (Abcam, Cat#ab22048), Ki67 (Cell Signaling Technology, Cat#9129), claudin-5 (Proteintech, Cat#29767-1-AP), NGP (ThermoFisher, Cat#orb420374), and caspase-3 (Proteintech, Cat#82202-1-PBS) were applied, followed by the appropriate secondary antibodies and DAPI.

## 16S rRNA sequencing analysis

DNA was extracted from mouse fecal samples using the TIANamp Bacteria DNA kit (Tiangen, Cat#4992448). High-throughput sequencing of the V4–V5 region of the bacterial 16S rRNA gene was performed by Guangdong Magigene Bio-technology Co., Ltd. using the Illumina NovaSeq 6000 platform with primers 515F (GTGCCAGCMGCCGCGGTAA) and A806R (GGACTACVSGGGTATCTAAT). Sequence processing followed library construction, with annotation and clustering performed using the UPARSE algorithm. Analysis of α-diversity, β-diversity, and differential species composition was based on the operational taxonomic unit (OTU) table.

## Liquid chromatography-mass spectrometry metabolomics analysis

Serum samples were analyzed for untargeted metabolomics by Shanghai Personal Biotechnology Co., Ltd. The analysis was performed using a Vanquish UHPLC System (Thermo Fisher Scientific), and metabolites were detected with an Orbitrap Exploris 120 (Thermo Fisher Scientific) using ESI ionization. Data were processed with ProteoWizard software, annotated through a mass spectrometry database, and analyzed using R software.

## Transcriptome sequencing

Transcriptome sequencing was performed by Guangdong Magigene Biotechnology Co., Ltd. Total RNA was extracted from renal tissues, and quality control assays were conducted. Sequencing was carried out on the Illumina HiSeq 2500 platform using the PE150 strategy.

## Correlation analysis

To explore the potential correlations between microbiota, metabolites, and gene expression, Pearson correlation analysis was conducted. Statistical significance was determined with a $P$ value threshold of less than 0.05. The analysis utilized R software.

## *In vitro* experiment

Neutrophils were isolated from mice. Bone marrow was flushed from femurs and tibias using RPMI 1640 with 10% fetal bovine serum (FBS) and filtered through a 70 µm nylon cell strainer. Neutrophils were isolated using the EasySep Mouse Neutro-phil Enrichment Kit (STEMCELL Technologies, Cat#19762) following the manufacturer's instructions. Neutrophils were cultured in RPMI 1640 with 10% FBS. For experiments, $2 \times 10^5$ neutrophils per well were plated in 12-well plates and treated with 10 µM 3-indoleacrylic acid (MCE, Cat#HY-W015273) for 12 hours.

Neutrophils were treated with culture medium containing siRNA at 100 nmol/L; siRNA targeting neutrophilic granule protein (*Ngp*) (ThermoFisher, Cat#1320001) and negative

control siRNA (ThermoFisher, Cat#12935200) were used. After 24 hours, the medium was replaced with normal medium, and cells were treated with 10 µM 3-indoleacrylic acid for another 12 hours.

Mouse renal tubular epithelial cells (TCMK-1) were cultured in MEM medium with 10% FBS. TCMK-1 cells ($2 \times 10^5$) were seeded into 12-well plates and co-cultured with pretreated neutrophils at a 1:1 ratio for 48 hours, with neutrophils in inserts with 0.4-micron pores and TCMK-1 cells in the lower chamber.

## Western blot

Proteins were separated on a Mini-PROTEAN TGX Gel (Bio-Rad, Cat#4561093) and transferred to a membrane using the Trans-Blot Turbo system (Bio-Rad, Cat#1704156). Membranes were blocked and incubated with primary antibodies against NGP (1:1,000, ThermoFisher, Cat#600-401-GW9), β-actin (1:1,000, Proteintech, Cat#66009-1-Ig), claudin-5 (1:1,000, Proteintech, Cat#29767-1-AP), and GAPDH (1:1,000, Proteintech, Cat#60004-1-Ig). After washing, membranes were incubated with HRP-conjugated secondary antibodies and detected using the ChemiDocTM MP Imaging System (Bio-Rad).

## qRT-PCR analysis

Total RNA was isolated using the RNeasy Kit (Qiagen, Cat#74104). Reverse transcription was performed with the Transcriptor First Strand cDNA Synthesis Kit (Roche, Cat#04379012001). The cDNA was analyzed using SsoAdvanced Universal SYBR Green Supermix (Bio-Rad, Cat#172-5270).

## Cell counting kit-8 assay

Cells were washed with PBS, and cell counting kit-8 (CCK-8) solution was added post-stimulation. Absorbance at 450 nm was measured with a spectrophotometer.

## Statistical analysis

Data are expressed as mean ± standard deviation. Statistical analyses were performed using the Student's *t*-test for comparing two groups, one-way ANOVA for multiple group comparisons of parametric data, or the Wilcoxon matched-pairs signed-ranks test for non-parametric data. A *P*-value of less than 0.05 was considered to indicate statistical significance.

## RESULTS

### Oral infection with *P. gingivalis* exacerbated kidney and intestinal injury in AKI

Emerging evidence points to a crucial involvement of gut microbiota in the development of AKI (22). In our experiments, mice in the AKI + *P. gingivalis* group displayed an increased kidney mass ratio (KW/BW) and significantly elevated levels of blood urea nitrogen, creatinine, and uric acid, as detailed in Table S1. HE and Masson's trichrome staining of kidney tissues showed that oral infection with *P. gingivalis* aggravated AKI-associated renal damage (Fig. 1A and B). When Abx treatment was administered to the AKI + *P. gingivalis* group, kidney function improved significantly (Table S1). This improvement was accompanied by reduced renal damage (Fig. 1A and B), suggesting a pivotal role of microbiota in the worsening of AKI by *P. gingivalis*.

Further histological examinations of intestinal tissues revealed notable epithelial damage and desquamation, along with lymphocytic infiltration, which were more pronounced in the AKI + *P. gingivalis* group (Fig. 1C). Toll-like receptor 4 (TLR4), which recognizes pathogen-associated molecular patterns such as the lipopolysaccharide from *P. gingivalis* (23), was significantly upregulated in the intestinal tissues of this group. Immunohistochemical staining confirmed this increased expression of TLR4. Notably,

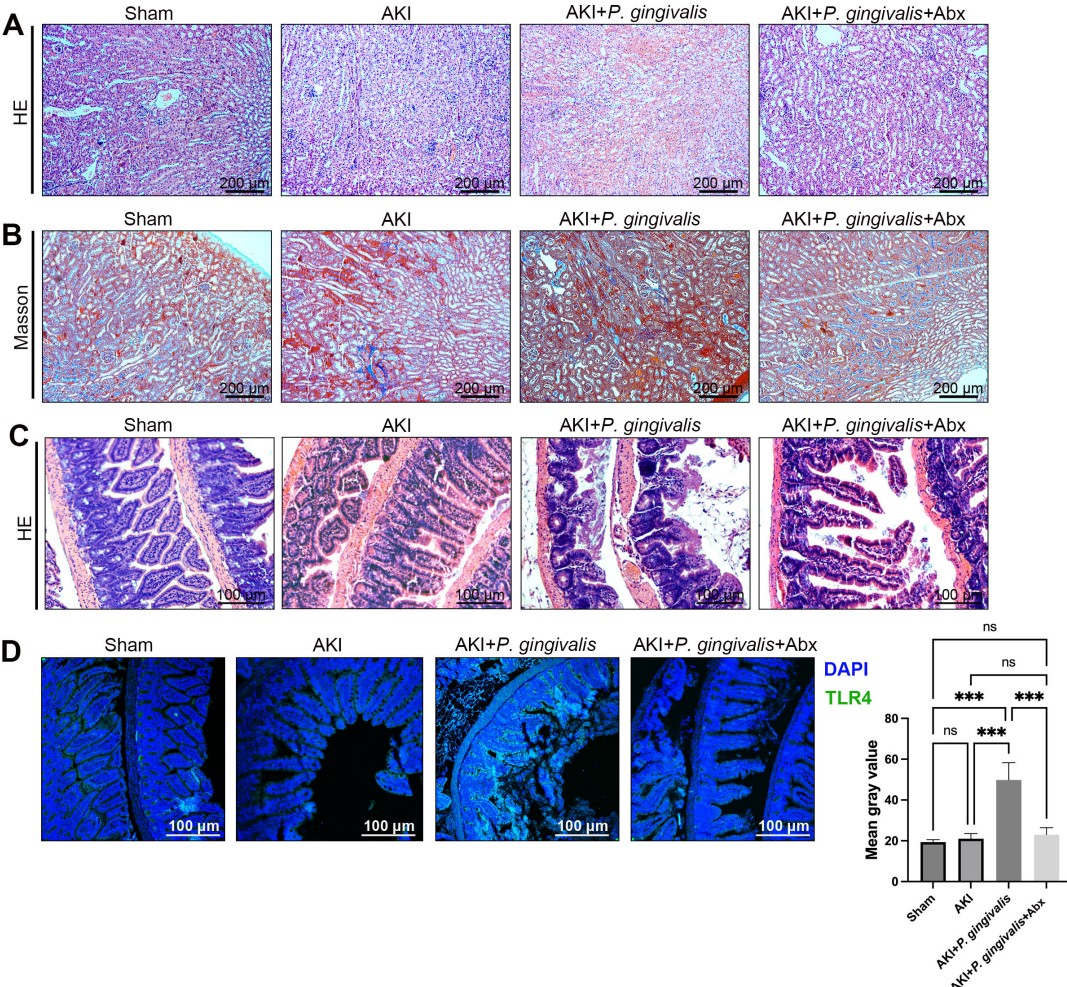

**FIG 1** Exacerbation of kidney and intestinal injury in AKI mice with *P. gingivalis* oral infection. (A) HE staining of renal tissue. (B) Masson's trichrome staining of renal tissue. (C) HE staining of intestinal tissue. (D) Immunofluorescence staining for TLR4 in intestinal tissue and quantitative analysis. The results are expressed as the mean ± SD. *$P < 0.05$; **$P < 0.01$; ***$P < 0.001$; and ****$P < 0.0001$ by ANOVA.

Abx treatment mitigated the intestinal damage and decreased TLR4 expression in these mice (Fig. 1D).

## Impact of oral infection with *P. gingivalis* on gut microbiota composition in AKI mice

To further investigate the effects of oral infection with *P. gingivalis* on gut microbiota composition in AKI, we conducted 16S rRNA sequencing analysis on fecal samples. This analysis revealed 578 identical operational taxonomic units shared among the four experimental groups, with each group also harboring unique OTUs (Fig. 2A). Notably, the most prevalent OTU remained consistent between the Sham and AKI groups but differed in the AKI + *P. gingivalis* and Abx treatment groups (Fig. 2B). The α-diversity analysis did not reveal any statistically significant differences among the groups (Fig. 2C). While AKI alone (without *P. gingivalis* infection) did not significantly alter the β-diversity of the gut microbiota compared to the Sham group, oral infection with *P. gingivalis* markedly changed the β-diversity within the AKI mice, indicating significant deviations from both the AKI and Sham groups. The Abx treatment also modified the gut microbiota composition in the AKI + *P. gingivalis* group, although these changes still varied from the Sham baseline (Fig. 2D).

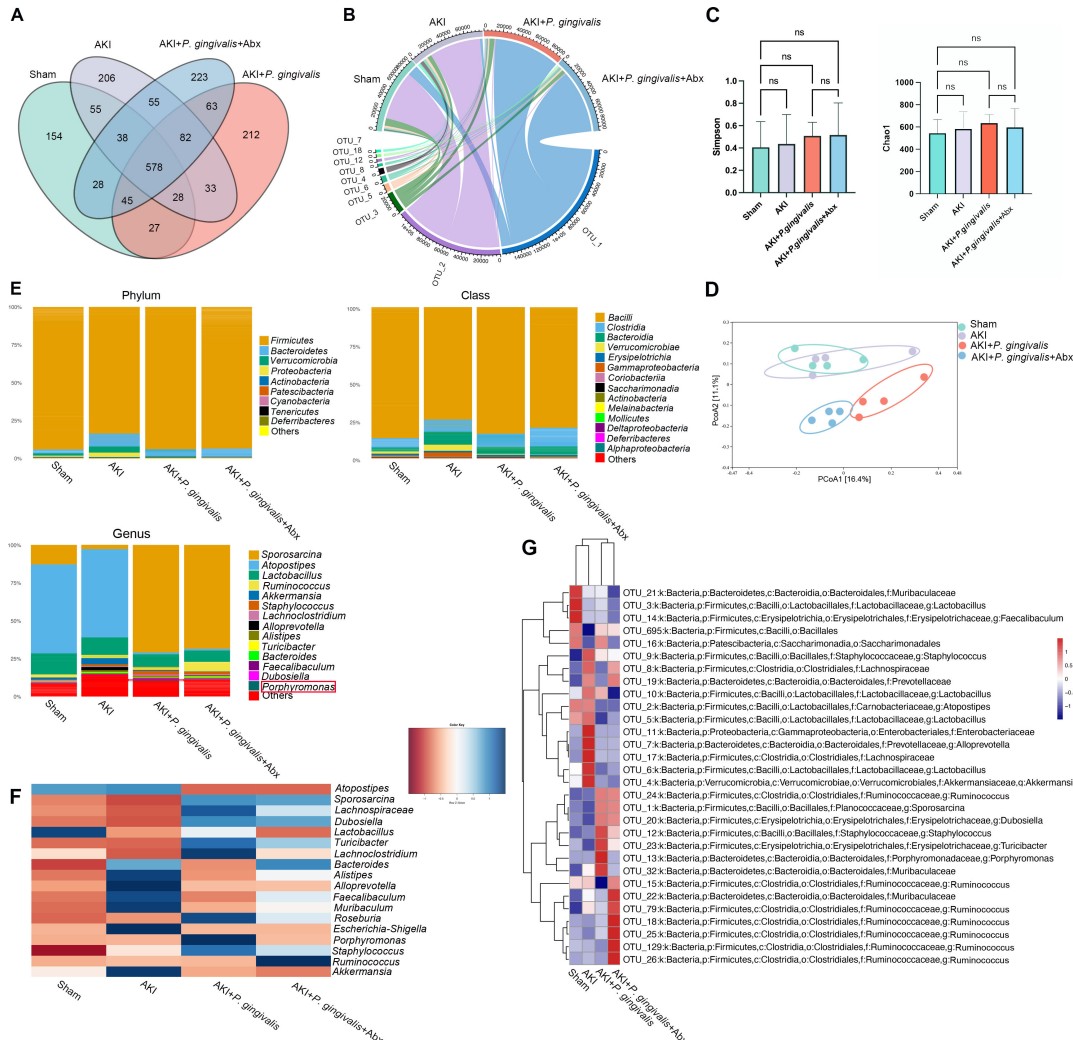

**FIG 2** Oral infection with *P. gingivalis* altered the gut microbiota composition in AKI mice. (A) The number of shared and unique operational taxonomic units among the various groups. (B) The OTU composition of microbial communities across the groups. (C and D) Analysis of the diversity and structure of the gut microbiota within each group. The α-diversity was presented by Chao1 and Simpson indices. ns, not significant; *P < 0.05; **P < 0.01; ***P < 0.001; ****P < 0.0001, determined using the one-way ANOVA test with Scheffé's *post hoc* analysis. The β-diversity was presented as a principal coordinate analysis (PCoA) plot based on UniFrac distances. (E) The percentages of the major phyla, classes, and genera within the gut microbiota. (F) The relative abundance of 18 major genera across the groups. (G) The clustering analysis of the top 30 OTUs with the highest abundances.

Further analysis of the gut microbiota composition at various taxonomic levels showed that *Firmicutes* was the most abundant phylum across all groups. At the class level, *Bacilli* dominated in all groups. However, noticeable differences emerged at the genus level: *Atopostipes* was most abundant in the Sham and AKI groups, whereas *Sporosarcina* was more prevalent in the AKI + *P. gingivalis* and Abx treatment groups (Fig. 2E). Examination of the top 18 most abundant genera revealed that the AKI + *P. gingivalis* group exhibited uniquely high abundances of *Porphyromonas*, *Staphylococcus*, and *Lachnospiraceae*. Post-Abx treatment, the abundance of these microorganisms notably decreased (Fig. 2F; Fig. S2A). A heatmap further illustrated the clustering analysis of the top 30 OTUs by abundance (Fig. 2G). Phylogenetic tree analysis underscored the similarities between the AKI and Sham groups, as well as between the AKI + *P. gingivalis* and Abx treatment groups (Fig. S2B). Additionally, KEGG pathway prediction based on the 16S rRNA sequencing data were conducted for each group, providing insights into potential metabolic functions (Fig. S2C).

In summary, our analysis confirmed that oral infection with *P. gingivalis* significantly impacted the gut microbiota. The notable increase in microbes in the AKI + *P. gingivalis* group might play a critical role in exacerbating AKI.

## *P. gingivalis*-induced metabolic alterations in AKI serum and the further elevation of 3-indoleacrylic acid

To further investigate how oral infection with *P. gingivalis* impacts gut microbiota and subsequently affects disease progression through metabolism, we conducted non-targeted metabolomics sequencing on serum samples from mice. The sequencing results revealed a substantial number of differential metabolites among the groups. To gain a clearer picture of the overall metabolite profile, we performed a principal component analysis on the metabolites. The findings indicated significant differences in the composition of serum metabolites among the Sham, AKI, and AKI + *P. gingivalis* groups in both the positive and negative ion modes (Fig. S3A). Under positive ion mode, there were 666 common differential metabolites shared among the Sham, AKI, and AKI + *P. gingivalis* groups, while under negative ion mode, this number rose to 1,186 (Fig. 3A). A meticulous examination of the differential metabolites revealed a balanced distribution of upregulated and downregulated metabolites in the AKI + *P. gingivalis* group compared to the AKI group. However, when compared to the Sham group, both the AKI + *P. gingivalis* and AKI groups exhibited a higher prevalence of upregulated metabolites (Fig. 3B). Cluster analysis further distinguished the serum metabolite compositions of the AKI + *P. gingivalis* and AKI groups from that of the Sham group, indicating a significant divergence (Fig. 3C).

To pinpoint the metabolic distinctions between the AKI and Sham groups, a volcano plot analysis was employed. This analysis highlighted an upregulation of metabolites such as L-isoleucine, quin, 3-indoleacrylic acid, stearic acid, 4-hydroxyphenylpyruvic acid, and phenyl sulfate in AKI mice, in contrast to the downregulation of pyroglutamic acid, deoxycholic acid, mycophenolic acid, adenine, and sorbitol (Fig. 3D). A comparative analysis between the AKI + *P. gingivalis* and AKI groups unveiled several significant alterations, with metabolites like pyroglutamic acid, deoxycholic acid, chenodeoxycholic acid, mycophenolic acid, 3-indoleacrylic acid, and sorbitol showing an increase, and alpha-dimorphecolic acid, prostaglandin-c2, stearic acid, ursodeoxycholic acid, and urocanic acid showing a decrease in the AKI + *P. gingivalis* group (Fig. 3D). Notably, the alterations in 3-indoleacrylic acid were particularly pronounced, with a marked increase in AKI mice and an even more substantial rise post-*P. gingivalis* infection (Fig. S3B).

A KEGG pathway enrichment analysis of the differential metabolites between the AKI + *P. gingivalis* and AKI groups identified enrichment in pathways, including oxidative phosphorylation, intestinal immune network for IgA production, and Th17 cell differentiation, underscoring the profound impact of *P. gingivalis* infection (Fig. S3C). Our study not only uncovered metabolic differences between AKI mice and normal mice but also demonstrated the profound impact of oral infection with *P. gingivalis* on the serum metabolites of AKI mice, notably leading to a significant upregulation of metabolites like 3-indoleacrylic acid.

## Multi-omics analysis discovered that oral infection with *P. gingivalis* induced gut-kidney axis dysregulation

Oral infection with *P. gingivalis* disrupted intestinal homeostasis in AKI mice. An in-depth analysis revealed a significant association between the top 14 most abundant gut microbiota and renal function, including genera such as *Porphyromonas*, *Dubosiella*, *Staphylococcus*, and *Lachnospiraceae* (Fig. 4A). We also noted key metabolites, especially 3-indoleacrylic acid, exhibiting the most potent correlation with renal function (Fig. 4B). Further investigation indicated a significant positive correlation between the increase of 3-indoleacrylic acid and the abundance of *Porphyromonas* in the comparison between the AKI and AKI + *P. gingivalis* groups (Fig. 4C).

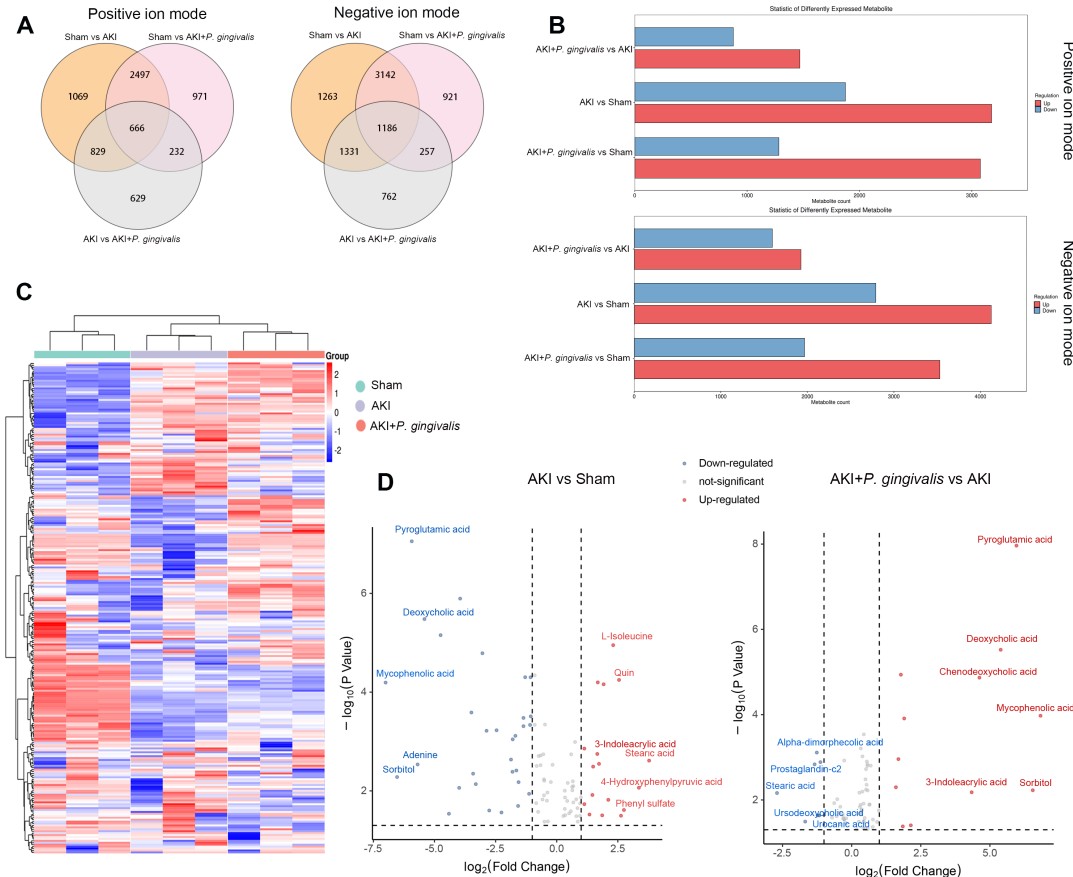

**FIG 3** Impact of oral infection with *P. gingivalis* on metabolic alterations in the serum of AKI mice. (A) The Venn diagram illustrates the number of shared and unique metabolites among the various groups. The left panel represents positive ion modes, while the right panel depicts negative ion modes. (B) Bar charts summarize the differential metabolites between each pair of groups. The upper section displays positive ion modes, and the lower section displays negative ion modes. The *X*-axis represents the number of differential metabolites, while the *Y*-axis indicates the comparison groups. Red bars indicate upregulation, and blue bars indicate downregulation. (C) Overall metabolite clustering analysis and heatmap visualization. (D) The volcano plot displays the distribution and trends of differential metabolites in the serum.

Our previous research has shown notable changes in the kidney transcriptome of AKI mice following oral infection with *P. gingivalis*. We further analyzed the top 30 genes exhibiting the most significant differences between the AKI and AKI + *P. gingivalis* groups, finding a close correlation between inflammatory factors and renal function indicators, such as neutrophilic granule protein (*Ngp*), *Il6*, *Il1r2*, *Osm*, and *Cxcl2* (Fig. S4A), Moreover, the results of qPCR experiments verified those of the transcriptome sequencing (Fig. S4B). A comprehensive multi-omics analysis revealed that these inflammatory factors not only have a significant connection with changes in gut microbiota but also correlate with metabolic differences in the host's blood (Fig. 4D; Fig. S4C). This suggests that oral infection with *P. gingivalis* may affect host metabolites by altering the gut microbiome, subsequently regulating the transcription level of kidney cells in mice.

It is worth noting that 3-indoleacrylic acid increased under AKI conditions and further increased when AKI was exacerbated by oral infection with *P. gingivalis* (Fig. S3B). Additionally, it showed a significant positive correlation with the abundance of *Porphyromonas* in the intestine (Fig. 4C). Through KEGG pathway enrichment analysis, we found that 3-indoleacrylic acid was primarily associated with immune diseases and signaling molecule interactions (Fig. S4D). Incorporating 16S rRNA, metabolomics, and transcriptomics into network analysis, we identified a close, positive correlation among 3-indoleacrylic acid, *Porphyromonas*, and *Ngp* (Fig. 4E). In conclusion, these multi-omics analyses indicated that oral infection with *P. gingivalis* might exacerbate AKI via the

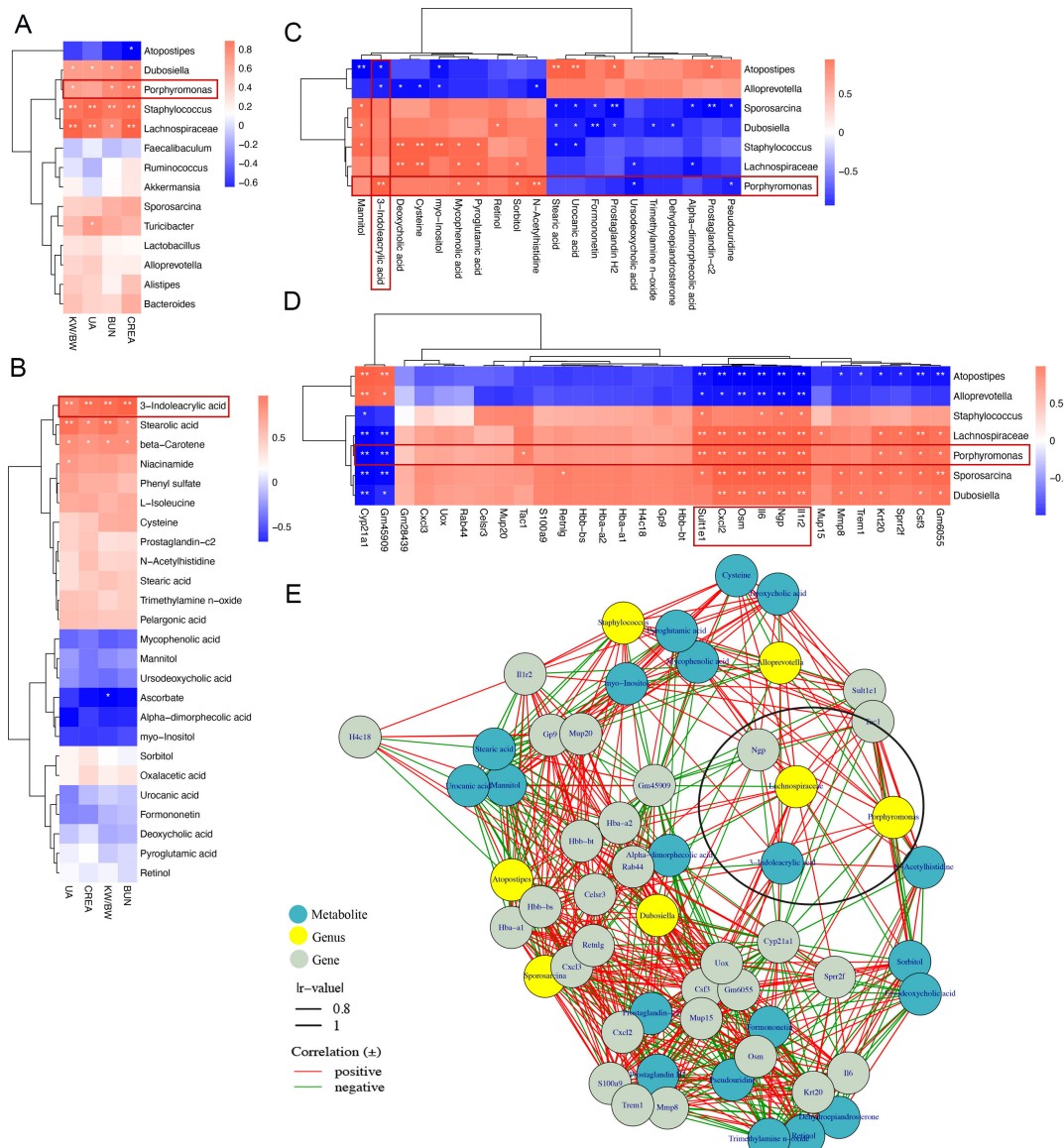

**FIG 4** Correlation analysis among gut microbiota, serum metabolites, and renal transcriptome in mice. (A) Heatmap representation of correlations between the most abundant gut microbiota and renal function parameters. (B) Heatmap visualization of the associations between major serum metabolites and renal function parameters. (C) Correlation analysis of major differential gut microbiota and top 20 differential serum metabolites between the AKI + *P. gingivalis* group and the AKI group. (D) Correlation analysis of major differential gut microbiota and top 30 differential renal tissue genes between the AKI + *P. gingivalis* group and the AKI group. (A–D) Spearman's correlation coefficients (*r* value) are depicted using a square color map. Asterisks indicate statistically significant correlation analysis results: *P < 0.05; **P < 0.01. (E) Correlation network diagram depicting the interactions among differential gut microbiota, differential serum metabolites, and differential renal tissue genes in the AKI + *P. gingivalis* group compared to the AKI group. Green dots represent metabolites, yellow dots represent gut microbiota, and gray dots represent genes.

gut-kidney axis, involving 3-indoleacrylic acid elevation and *Ngp* expression in the kidney.

## 3-Indoleacrylic acid mediated neutrophil-derived NGP production leading to renal tubular epithelial cell injury

We have identified a potential association between 3-indoleacrylic acid and the elevated expression of NGP in the kidney. In our *in vivo* experiments, high concentrations of 3-indoleacrylic acid were associated with a significant increase in NGP expression in the

AKI + *P. gingivalis* group compared to the other groups, with NGP expression predominantly localized near renal epithelial cells (Fig. S5A). Moreover, the expression of the epithelial junction protein CLDN5 was significantly reduced in the AKI + *P. gingivalis* group (Fig. S5B). Additionally, caspase-3 expression was markedly elevated in the AKI + *P. gingivalis* group relative to the other groups (Fig. S5C). Notably, the increase in NGP expression was closely accompanied by enhanced caspase-3 expression in kidney tissue, suggesting a potential role for NGP in promoting apoptosis during AKI (Fig. S6).

Further *in vitro* experiments indicated that the stimulation of 3-indoleacrylic acid led to increased NGP protein levels in mouse neutrophils (Fig. 5A and B). Concurrently, we also observed an elevation in the expression of inflammatory factors such as *Il6*, *Osm*, and *Cyp21a1* (Fig. 5C). It was noted that 3-indoleacrylic acid stimulation did not affect renal epithelial cell proliferation (Fig. 5D). However, the 3-indoleacrylic acid-pretreated neutrophils resulted in a reduction in the expression of Ki67, a protein related to nuclear division and proliferation in epithelial cells (Fig. 5E). Interestingly, when NGP expression in neutrophils was inhibited, the 3-indoleacrylic acid-pretreated neutrophils no longer affected the expression of Ki67 and CLDN5 in epithelial cells (Fig. 5E and F). These experimental results suggested that 3-indoleacrylic acid could stimulate neutrophils to express NGP, which in turn caused damage to renal epithelial cells.

## DISCUSSION

Periodontitis not only leads to the destruction and loss of periodontal support tissues but is also associated with various systemic diseases such as hypertension and diabetes. A substantial body of recent research has indicated a significant correlation between periodontitis and kidney disease, with the treatment of periodontitis improving kidney function (24, 25). However, current studies have primarily focused on chronic kidney disease (26), and the connection with AKI remains unclear. Our study discovered that oral infection with *P. gingivalis*, a key pathogen in periodontitis, exacerbated AKI damage and significantly disrupted the gut microbiota in AKI mice, leading to changes in the proportions of gut microbial communities, particularly an increase in *Porphyromonas*, *Staphylococcus*, and *Lachnospiraceae*. Additionally, there was a noticeable change in serum metabolites and an increase in inflammatory factors in kidney tissue.

The microbiota and its metabolites play a crucial role in shaping the immune system beyond the gut (27). In chronic kidney disease, dysbiosis of the gut microbiota and the accumulation of gut-derived uremic toxins can further exacerbate the condition (28). In a mouse model of renal ischemia-reperfusion injury, the gut microbiota has been shown to cause AKI, and the depletion of the gut microbiota can prevent such injury (21). Furthermore, the supplementation of specific short-chain fatty acids or the introduction of bacterial strains that produce these short-chain fatty acids can mitigate ischemic post-AKI in mice (29, 30). This suggests that the impact of the microbiota on the severity of AKI may be related to changes in bacterial strains that produce immunomodulatory metabolites. However, the influence of periodontitis on the gut microbiota and its metabolism in AKI, particularly the role of *P. gingivalis* oral infection, is not well understood. Our study found that after *P. gingivalis* oral infection, the gut microbiota changed, leading to a significant alteration in the host's metabolic products, especially a further increase in serum 3-indoleacrylic acid. This metabolite, a tryptophan derivative produced by gut bacteria, profoundly impacts host physiology, including maintaining the epithelial barrier and immune function. Studies have shown that certain anaerobic streptococci produce 3-indoleacrylic acid by breaking down tryptophan, which can regulate tumor ferroptosis and promote the development of colorectal cancer (31). Our research also found that the increase in 3-indoleacrylic acid caused by *P. gingivalis* oral infection stimulated the high expression of NGP in neutrophils, thereby affecting the integrity of renal tubular epithelium. The stimulation of 3-indoleacrylic acid alone did not significantly change the activity of renal tubular epithelium, further confirming that the severity of AKI is influenced by changes in bacterial strains that produce immunomodulatory metabolites. In addition, we also measured the serum levels of indole-3-acetic acid

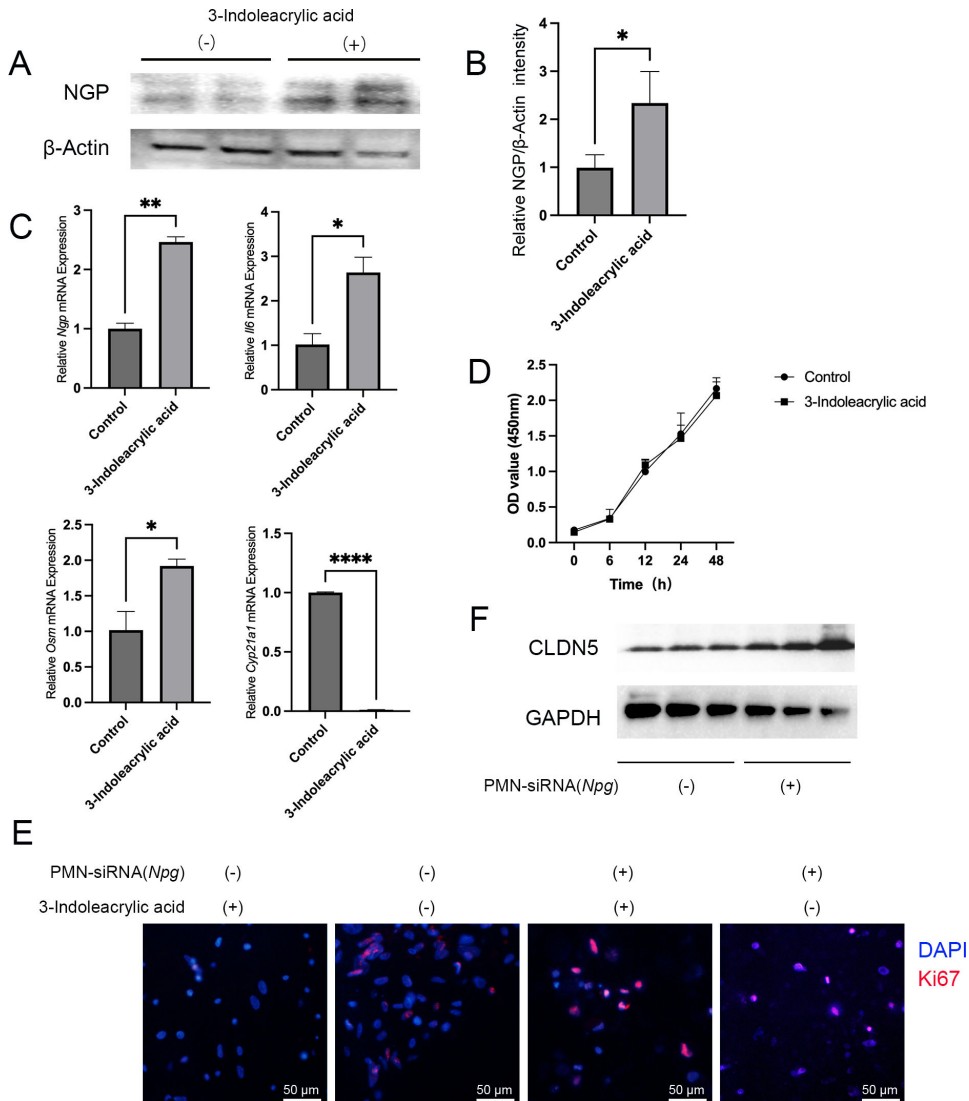

**FIG 5** Promotion of neutrophil-NGP by 3-indoleacrylic acid leading to renal epithelial cell injury. (A and B) Western blot analysis of NGP protein expression in neutrophils stimulated with 3-indoleacrylic acid, followed by quantitative analysis. (C) qRT-PCR analysis of *Ngp*, *Il6*, *Osm*, and *Cyp21a1* gene expression in neutrophils, normalized to *Gapdh*. (D) CCK-8 assay of renal epithelial cell viability after exposure to 3-indoleacrylic acid. (E) Ki67 immunohistochemical staining to assess the proliferative capacity of renal epithelial cells. (F) Western blot analysis of junctional proteins in renal epithelial cells co-cultured with neutrophils stimulated by 3-indoleacrylic acid after NGP inhibition. Results are expressed as mean ± SD. Statistical significance was determined using Student's *t*-test: *$P < 0.05$; **$P < 0.01$; ***$P < 0.001$; and ****$P < 0.0001$.

in mice. Indole-3-acetic acid is a metabolite that is involved in a variety of cellular processes and disease states. The results revealed that its levels were notably decreased in the AKI group in contrast to the Sham group. However, *P. gingivalis* infection did not have a significant impact on its levels (Fig. S7).

Neutrophils play an important role in kidney disease, including participating in inflammatory responses, promoting renal fibrosis, and affecting the repair of kidney damage (32). Single-cell sequencing analysis has revealed that NGP is expressed during neutrophil differentiation as part of the secondary granule genes and is involved in the maturation and functional acquisition of these cells (33). In addition, NGP plays an important role in inflammation and phagocytosis and is closely related to the host's defense mechanisms (34). Our transcriptome data analysis found that after *P. gingivalis* oral infection, the expression of NGP in kidney tissue increased and was related to

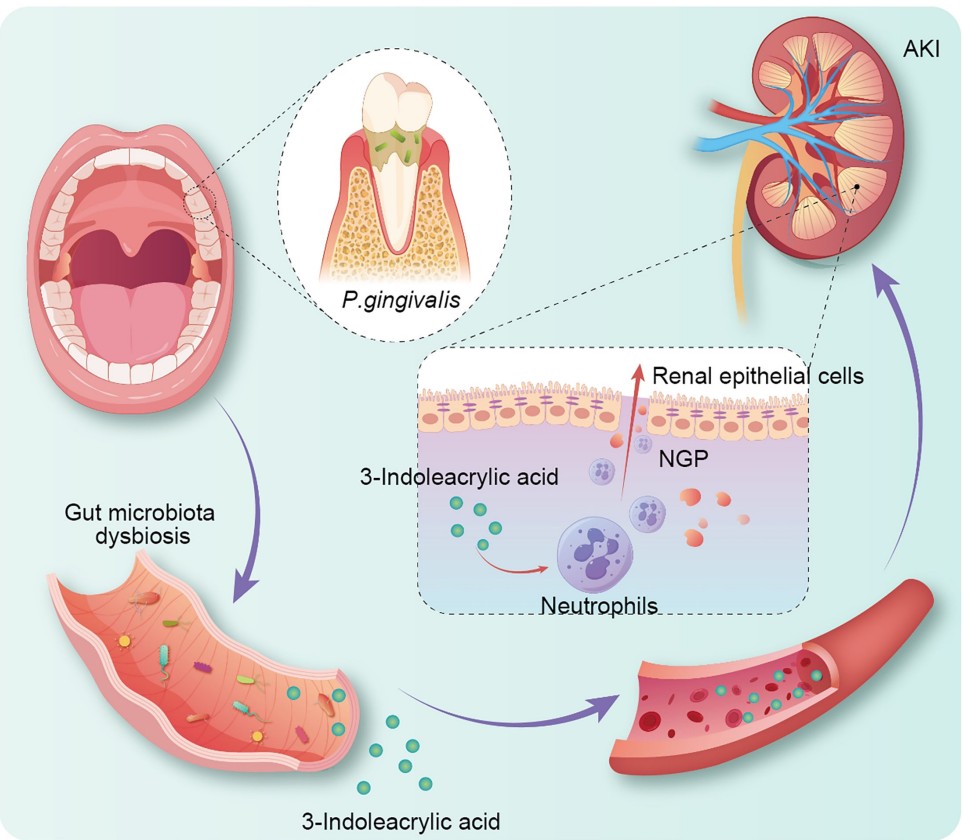

**FIG 6** Proposed mechanism diagram. *P. gingivalis* oral infection disrupted the balance of gut microbiota and was an important modifier determining the severity of AKI. *P. gingivalis* might cause an increase in the level of the gut microbial metabolite 3-indoleacrylic acid, interfering with kidney immunity and disrupting the maintenance of kidney epithelium.

the level of 3-indoleacrylic acid. Our cellular experiment results also suggested that 3-indoleacrylic acid affected the expression of NGP in neutrophils, and silencing NGP could inhibit the damaging effect of neutrophils on the renal tubular epithelium.

Regarding the potential guiding significance and application prospects for future clinical treatment, our study offers several important implications. In the aspect of diagnosis, the identified changes in gut microbiota, serum metabolites, and kidney tissue gene expression could potentially serve as biomarkers. For example, the level of 3-indoleacrylic acid in the serum might be used to assess the risk of AKI exacerbation related to *P. gingivalis* infection. In terms of treatment, intervening in the gut microbiota, either by using antibiotics as in our study or through other means like beneficial bacteria supplementation, could be a new strategy. Additionally, developing drugs that target the 3-indoleacrylic acid-mediated mechanism could provide new therapeutic options.

The gut microbiota-metabolic network was closely related to the development of AKI. *P. gingivalis* oral infection disrupted the balance of gut microbiota and was an important modifier determining the severity of AKI. Under the "gut-kidney axis," *P. gingivalis* might cause an increase in the level of the gut microbial metabolite 3-indoleacrylic acid, interfering with kidney immunity and disrupting the maintenance of kidney epithelium. Therefore, intervening in the gut microbiota may provide a new treatment strategy for AKI patients with periodontitis (Fig. 6).

## ACKNOWLEDGMENTS

This study was supported by the Young Scientist Program of Beijing Stomatological Hospital, Capital Medical University (YSP202314); the Second Batch of Science and

Technology Plan Projects of Jinan Municipal Health Commission (2020-3-49, 2023-2-161), the Dean's Research Fund of Jinan Stomatological Hospital (2021-01), and the Jinan Clinical Medical Science and Technology Innovation Plan Project (202328023).

W.W. and Q.J. conceived the study. L.D., W.W., and Z.J. performed laboratory assays and experiments. L.D. and J.S. analyzed the laboratory data. L.D. and J.H. produced the tables and figures. L.D. and W.W. wrote the first draft with assistance from Q.J. All authors critically reviewed and approved the final manuscript.

## AUTHOR AFFILIATIONS

[1]Beijing Stomatological Hospital, School of Stomatology, Capital Medical University, Beijing, China

[2]Department of Periodontology, Jinan Key Medical and Health Laboratory of Oral Diseases and Tissue Regeneration, Jinan Key Laboratory of Oral Diseases and Tissue Regeneration, Shandong Provincial Key Medical and Health Laboratory of Oral Diseases and Tissue Regeneration, Shandong Provincial Key Medical and Health Discipline of Oral Medicine, Jinan Stomatological Hospital, Jinan, Shandong, China

## AUTHOR ORCIDs

Wei Wei http://orcid.org/0000-0001-5874-3825

## FUNDING

| Funder | Grant(s) | Author(s) |
| --- | --- | --- |
| Young scientist program of Beijing stomatological hospital, Capital Medical University | YSP202314 | Wei Wei |
| the second batch of science and technology plan projects of jinan municipal health commission | 2020-3-49, 2023-2-161 | Jing Sun |
| Dean's research fund of jinan stomatological hospital | 2021-01 | Jing Sun |
| jinan clinical medical science and technology innocation plan project | 202328023 | Jing Sun |

## AUTHOR CONTRIBUTIONS

Ling Dong, Conceptualization, Supervision, Writing – review and editing, Data curation, Formal analysis, Investigation | Jiangqi Hu, Supervision, Data curation | Qingsong Jiang, Methodology, Project administration, Writing – original draft | Wei Wei, Methodology, Conceptualization, Supervision, Funding acquisition, Writing – review and editing, Data curation, Formal analysis, Resources, Project administration, Validation, Investigation.

## DATA AVAILABILITY

The raw sequence data reported in this paper have been deposited in the Genome Sequence Archive in National Genomics Data Center, China National Center for Bioinformation/Beijing Institute of Genomics, Chinese Academy of Sciences (https://ngdc.cncb.ac.cn/gsa: CRA017606 and CRA014139; https://ngdc.cncb.ac.cn/omix: OMIX006815).

## ETHICS APPROVAL

The study protocols were approved by the Institutional Animal Care and Use Committee of Capital Medical University.

## ADDITIONAL FILES

The following material is available online.

## Supplemental Material

**Supplemental material (mSystems01136-24-S0001.docx).** Fig. S1-S7, Table S1, and supplemental methods and materials.

## Open Peer Review

**PEER REVIEW HISTORY (review-history.pdf).** An accounting of the reviewer comments and feedback.

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
