## [Reviewer comments · mSystems]

Multi-Omics Investigation of *Porphyromonas gingivalis* Exacerbating Acute Kidney Injury through the Gut-Kidney Axis

Ling Dong, Zhaoxin Ji, Jing Sun, Jiangqi Hu, Qingsong Jiang, and WEI WEI

Corresponding Author(s): WEI WEI, Department of Prosthodontics, Beijing Stomatological Hospital, School of Stomatology, Capital Medical University

Review Timeline:

Submission Date:	August 21, 2024
Editorial Decision:	November 11, 2024
Revision Received:	December 4, 2024
Accepted:	December 13, 2024

Editor: Aleksandra Nita-Lazar

Reviewer(s): Disclosure of reviewer identity is with reference to reviewer comments included in decision letter(s). The following individuals involved in review of your submission have agreed to reveal their identity: Ran Zhang (Reviewer #1)

Transaction Report:

DOI: <https://doi.org/10.1128/mSystems.01136-24>

Re: mSystems01136-24 (Multi-Omics Investigation of Porphyromonas gingivalis Exacerbating Acute Kidney Injury through the Gut-Kidney Axis)

Dear Dr. Jing Sun:

Revision Guidelines

Sincerely,
Aleksandra Nita-Lazar
Editor
mSystems

Reviewer #1 (Comments for the Author):

This manuscript presents a comprehensive and well-designed study on the relationship between Porphyromonas gingivalis oral infection and acute kidney injury (AKI) through the gut-kidney axis. The multi-omics approach used provides a detailed understanding of the underlying mechanisms. The results are clearly presented and support the conclusions drawn. However, the following suggestions may further enhance the manuscript:

1. In the analysis of kidney tissue, it could be considered to add immunohistochemical staining for NGP expression in kidney tissue to more intuitively demonstrate its expression in the tissue.
2. Further perform immunofluorescence staining for cldn5 in kidney tissue to better observe its localization and expression changes in renal epithelial cells.
3. In addition to the existing transcriptome sequencing analysis, qPCR expression detection of some key genes in kidney tissue could be added to verify the accuracy of the transcriptome data.
4. Indole-3-acetic acid is a common metabolite related to diseases. How does it change? This aspect could be explored and discussed in the manuscript.
5. It is recommended to elaborate more on the selection basis of experimental animals in the article. For example, explain why C57BL/6J mice were chosen and the advantages and limitations of this strain in the study.
6. It is suggested to further discuss the similarities and differences between the results of this study and those of other related studies, as well as the potential guiding significance and application prospects of this study for future clinical treatment.

Reviewer #2 (Comments for the Author):

1. The authors have conducted an in-depth investigation into the impact of *P. gingivalis* oral infection on AKI through the gut-kidney axis. However, to further enhance our understanding of the pathophysiological process, it would be highly beneficial to explore the apoptotic status of kidney tissue. Utilize caspase - 3 detection to assess the degree of apoptosis in kidney tissue during the process of AKI induced by *P. gingivalis* oral infection. This will provide direct evidence of the extent of cell death and its relationship with the pathogen and AKI development.
2. Given the observed changes in NGP expression, is there a connection between kidney tissue apoptosis and NGP?
3. The integrity of the renal epithelium is of great importance in AKI. Therefore, it is essential to study the expression changes of epithelial junction proteins.

To editors:

The study on the gut-kidney axis and *P. gingivalis*-induced AKI using a multi-omics approach is cutting-edge. It has broad implications for research and clinical practice, justifying publication in mSystems. The supplemental material make a substantive enough contribution to the work to warrant posting to mSystems.

To authors:

1. The authors have conducted an in-depth investigation into the impact of *P. gingivalis* oral infection on AKI through the gut-kidney axis. However, to further enhance our understanding of the pathophysiological process, it would be highly beneficial to explore the apoptotic status of kidney tissue. Utilize caspase - 3 detection to assess the degree of apoptosis in kidney tissue during the process of AKI induced by *P. gingivalis* oral infection. This will provide direct evidence of the extent of cell death and its relationship with the pathogen and AKI development.
2. Given the observed changes in NGP expression, is there a connection between kidney tissue apoptosis and NGP?
3. The integrity of the renal epithelium is of great importance in AKI. Therefore, it is essential to study the expression changes of epithelial junction proteins.

Point-by-point responses to the reviewers' comments and criticisms:

Editor's Remarks to Author:

Thank you for the privilege of reviewing your work. Below you will find my comments, instructions from the mSystems editorial office, and the reviewer comments. Please return the manuscript within 60 days; if you cannot complete the modification within this time period, please contact me. If you do not wish to modify the manuscript and prefer to submit it to another journal, notify me immediately so that the manuscript may be formally withdrawn from consideration by mSystems.

Response: We sincerely thank the Editor for the feedback and the opportunity to resubmit our revised manuscript. We have carefully considered the expert reviewers' comments and implemented comprehensive changes to address their concerns. We believe these revisions have significantly enhanced both the quality and clarity of our manuscript.

Revision Guidelines

- Upload point-by-point responses to the issues raised by the reviewers in a file named "Response to Reviewers," NOT in your cover letter.
- Upload a compare copy of the manuscript (without figures) as a "Marked-Up Manuscript" file.

- Upload a clean .DOC/.DOCX version of the revised manuscript and remove the previous version.
- Each figure must be uploaded as a separate, editable, high-resolution file (TIFF or EPS preferred), and any multipanel figures must be assembled into one file.
- Any supplemental material intended for posting by ASM should be uploaded with their legends separate from the main manuscript. You can combine all supplemental material into one file (preferred) or split it into a maximum of 10 files with all associated legends included.

For complete guidelines on revision requirements, see our Submission and Review Process webpage. Submission of a paper that does not conform to guidelines may delay acceptance of your manuscript.

Response: We appreciate the Editor's efforts in highlighting these important issues. As requested, we have prepared a separate file titled "Response to Reviewers" in accordance with the journal's requirements. Additionally, we have thoroughly reviewed our manuscript to ensure that it complies fully with the revision guidelines and journal standards.

Response: We sincerely thank the Editor for the reminder. We have verified all links to sequence records and ensured that each accession number retrieves the full record of the data. Regarding publication fees and ASM membership, we are aware of the requirements and will ensure timely payment as per the journal's policy.

Reviewer comments:

Reviewer #1 (Comments for the Author):

This manuscript presents a comprehensive and well-designed study on the relationship between *Porphyromonas gingivalis* oral infection and acute kidney injury (AKI) through the gut-kidney axis. The multi-omics approach used provides a detailed understanding of the underlying mechanisms. The results are clearly presented and support the conclusions drawn. However, the following suggestions may further enhance the manuscript:

Response: We sincerely thank Reviewer #1 for recognizing the comprehensiveness and design of our study. We greatly appreciate the valuable suggestions, which have been instrumental in further improving the quality of our manuscript. We have carefully addressed each of the Reviewer's comments and incorporated the necessary revisions into the manuscript.

1. In the analysis of kidney tissue, it could be considered to add immunohistochemical staining for NGP expression in kidney tissue to more intuitively demonstrate its expression in the tissue.

Response: We sincerely thank the Reviewer for highlighting this important point. In response, we have performed immunohistochemical staining on kidney tissues from each group, as suggested. The results indicate that NGP expression was significantly higher in the AKI + *P. gingivalis* group compared to other groups, with the expression localized near renal epithelial cells. These findings suggest that NGP may be associated with renal epithelial cell damage. We have included these results in the revised manuscript. Please see Supplementary Figure 5A and refer to the updated text (Lines 361-365, Page 16).

2. Further perform immunofluorescence staining for *cldn5* in kidney tissue to better observe its localization and expression changes in renal epithelial cells.

Response: We sincerely thank the Reviewer for this insightful suggestion. In response, we conducted immunofluorescence staining for CLDN5 in kidney tissue to observe its localization and expression changes in renal epithelial cells. The results revealed that the expression level of CLDN5 was significantly lower in the AKI + *P. gingivalis* group compared to the other groups, consistent with the observed trend of kidney damage. These detailed results and the corresponding images have been included in the revised manuscript. Please refer to Supplementary Figure 5B and the relevant section (Lines 365-366, Page 16).

3. In addition to the existing transcriptome sequencing analysis, qPCR expression detection of some key genes in kidney tissue could be added to verify the accuracy of the transcriptome data.

Response: We sincerely thank the Reviewer for this valuable recommendation. In response, we performed qPCR analysis to detect the expression of several key genes in kidney tissue to validate the transcriptome sequencing results. The qPCR findings were highly consistent with the transcriptome data, thereby confirming the accuracy and reliability of our sequencing analysis. These qPCR results have been incorporated into the revised manuscript. Please refer to Supplementary Figure 4B and the relevant section (Lines 338-339, Page 15).

4. Indole-3-acetic acid is a common metabolite related to diseases. How does it change? This aspect could be explored and discussed in the manuscript.

Response: We sincerely thank the Reviewer for raising this insightful question. Indole-3-acetic acid is a metabolite implicated in various cellular processes and disease states (Li H, et al. Gut microbiota-derived indole-3-acetic acid suppresses high myopia progression by promoting type I collagen synthesis. *Cell Discov* 2024. 10: 89; Yi Zhou, et al. The role of the indoles in microbiota-gut-brain axis and potential therapeutic targets: A focus on human neurological and neuropsychiatric diseases. *Neuropharmacology* 2023. 239: 109690). In response, we measured serum levels of Indole-3-acetic acid in mice. The results show that its levels were significantly lower in the AKI group compared to the Sham group; however, *P. gingivalis* infection did not significantly alter its serum levels. We have included these findings in the revised manuscript. Please refer to Supplementary Figure 7 and the relevant section (Lines 423-428, Page 18) for details.

5. It is recommended to elaborate more on the selection basis of experimental animals in the article. For example, explain why C57BL/6J mice were chosen and the advantages and limitations of this strain in the study.

Response: We appreciate the reviewer's inquiry regarding our choice of the C57BL/6 mouse strain. Our selection was influenced by the extensive use of C57BL/6 mice in a variety of physiological and pathological studies, as documented in the AKI-related literature (Yao W, et al. Single Cell RNA Sequencing Identifies a Unique Inflammatory Macrophage Subset as a Druggable Target for Alleviating Acute Kidney Injury. *Adv Sci* 2022; 9: e2103675; Chen C, et al. Legumain promotes tubular ferroptosis by facilitating chaperone-mediated autophagy of GPX4 in AKI. *Cell Death Dis* 2021; 12: 65; Livingston MJ, et al. Tubular cells produce FGF2 via autophagy after acute kidney injury leading to fibroblast activation and renal fibrosis. *Autophagy*

2023; 19: 256-277). The C57BL/6 strain exhibits a pronounced AKI phenotype following surgery, making it a highly suitable model for our study. Using this well-characterized strain ensures reliable results and facilitates comparisons with prior research in the field.

The ischemia-reperfusion model, which we employed, offers the advantage of a short modeling cycle and a high success rate, making it widely applicable in studies on AKI pathogenesis and drug efficacy evaluation. While various rodent strains are used for AKI modeling (e.g., BALB/c, C57BL/6N, C57BL/6J, and Wistar rats), each strain exhibits different tolerances to surgical procedures, which should be carefully considered when designing experiments. We also acknowledge the potential value of expanding research to other animal models, such as experimental miniature pigs, whose renal tissue structure closely resembles that of humans.

We have added a detailed description of our rationale for selecting C57BL/6J mice to the revised manuscript. Please refer to the relevant section (Lines 109-112, Page 5-6).

6. It is suggested to further discuss the similarities and differences between the results of this study and those of other related studies, as well as the potential guiding significance and application prospects of this study for future clinical treatment.

Response: We are deeply grateful to the reviewer for the insightful comments and invaluable suggestions. We have thoroughly compared and discussed the similarities and differences between our study results and those of other related studies. In terms of similarities, like other studies, we also recognize the significant role of the gut microbiota in AKI. However, our study stands out with its comprehensive multi-omics approach. We have specifically identified the detailed changes in gut microbiota composition after *P. gingivalis* oral infection, such as the increase in certain genera,

and the unique metabolic alterations, like the elevation of 3-Indoleacrylic acid and its correlation with *Porphyromonas*.

Regarding the potential guiding significance and application prospects for future clinical treatment, our study offers several important implications. In the aspect of diagnosis, the identified changes in gut microbiota, serum metabolites, and kidney tissue gene expression could potentially serve as biomarkers. For example, the level of 3-Indoleacrylic acid in the serum might be used to assess the risk of AKI exacerbation related to *P. gingivalis* infection. In terms of treatment, intervening in the gut microbiota, either by using antibiotics as in our study or through other means like beneficial bacteria supplementation, could be a new strategy. Additionally, developing drugs that target the 3-Indoleacrylic acid-mediated mechanism could provide new therapeutic options. Moreover, our study highlights the importance of treating periodontitis in AKI patients to prevent further kidney damage.

We have incorporated an in-depth discussion of these aspects into the revised manuscript, highlighting how our research could contribute to the improvement of clinical management of AKI. Please refer to the relevant sections (Lines 407-413, Page 17-18).

Reviewer #2 (Comments for the Author):

1. The authors have conducted an in-depth investigation into the impact of *P. gingivalis* oral infection on AKI through the gut-kidney axis. However, to further enhance our understanding of the pathophysiological process, it would be highly beneficial to explore the apoptotic status of kidney tissue. Utilize caspase - 3 detection to assess the degree of apoptosis in kidney tissue during the process of AKI induced by *P. gingivalis* oral infection. This will provide direct evidence of the extent of cell death and its relationship with the pathogen and AKI development.

Response: We sincerely thank the Reviewer for the suggestion to examine the apoptotic status of kidney tissue using caspase-3 detection. In response, we performed immunohistochemical analysis to detect caspase-3 expression in kidney tissues from different groups. The results indicate significantly increased caspase-3 expression in the AKI + *P. gingivalis* group compared to other groups, suggesting that *P. gingivalis* oral infection promotes apoptosis in AKI kidney tissue. These results and the corresponding analysis have been included in the revised manuscript. Please see Supplementary Figure 5C and refer to the relevant section (Lines 367-368, Page 16).

2. Given the observed changes in NGP expression, is there a connection between kidney tissue apoptosis and NGP?

Response: We sincerely thank the Reviewer for the insightful comment regarding the connection between kidney tissue apoptosis and NGP. In our study, we observed that the increase in NGP expression was accompanied by enhanced caspase-3 expression in kidney tissue. These findings suggest that NGP may play a role in promoting apoptosis during AKI. We have included these results in the revised manuscript. Please refer to the relevant section (Lines 368-371, Page 16, Supplementary Figure 6).

3. The integrity of the renal epithelium is of great importance in AKI. Therefore, it is essential to study the expression changes of epithelial junction proteins.

Response: We sincerely thank the Reviewer for this valuable comment. We fully agree that the integrity of the renal epithelium plays a critical role in AKI and that examining the expression changes of epithelial junction proteins is essential. In our study, we specifically analyzed the expression of the epithelial junction protein CLDN5 and found that its expression was significantly decreased in kidney tissue following *P. gingivalis* oral infection. These findings highlight the potential involvement of impaired epithelial junctions in AKI pathogenesis. We have incorporated these results into the revised manuscript. Please refer to the relevant section (Lines 365-366, Page 16) and Supplementary Figure 5B for further details.

Re: mSystems01136-24R1 (Multi-Omics Investigation of Porphyromonas gingivalis Exacerbating Acute Kidney Injury through the Gut-Kidney Axis)

Dear Dr. WEI WEI:

Your manuscript has been accepted, and I am forwarding it to the ASM production staff for publication. Your paper will first be checked to make sure all elements meet the technical requirements. ASM staff will contact you if anything needs to be revised before copyediting and production can begin. Otherwise, you will be notified when your proofs are ready to be viewed.

Sincerely,
Aleksandra Nita-Lazar
Editor
mSystems

Reviewer #1 (Comments for the Author):

I am satisfied with the revisions made by the authors. They have comprehensively addressed all my previous concerns. The addition of multiple experimental results has significantly enhanced the manuscript's quality. The discussions on various aspects have also added great value. I believe the manuscript is worthy of publication.

Reviewer #2 (Comments for the Author):

I am satisfied with the revisions made by the authors in response to my previous comments. The manuscript has been significantly improved, and the concerns I raised have been adequately addressed.

Comments and Suggestions for the Author: I am satisfied with the revisions made by the authors in response to my previous comments. The manuscript has been significantly improved, and the concerns I raised have been adequately addressed.

Confidential remarks for the Editors: The authors' responses to the comments are comprehensive. I recommend that the manuscript be accepted for publication in mSystems. The study makes a contribution to the understanding of the relationship between *P. gingivalis* oral infection and AKI through the gut - kidney axis, and the revisions have addressed all my concerns raised during the review process.